**Brief Communication**

# A large-scale in silico replication of ecological and evolutionary studies

**Yefeng Yang** ⬤[1,5] ✉, **Erik van Zwet** ⬤[2,5], **Nikolaos Ignatiadis** ⬤[3] & **Shinichi Nakagawa** ⬤[1,4] ✉

Despite the growing concerns about the replicability of ecological and evolutionary studies, no results exist from a field-wide replication project. We conduct a large-scale in silico replication project, leveraging cutting-edge statistical methodologies. Replicability is 30%–40% for studies with marginal statistical significance in the absence of selective reporting, whereas the replicability of studies presenting 'strong' evidence against the null hypothesis $H_0$ is >70%. The former requires a sevenfold larger sample size to reach the latter's replicability. We call for a change in planning, conducting and publishing research towards a transparent, credible and replicable ecology and evolution.

The rapidly evolving field of meta-science has spotlighted that many published research findings have low credibility[1–3]. Large-scale collaborative replication projects in the social and natural sciences, such as the Open Science Collaboration[3] and Registered Replication Reports[4] reveal a low replication probability of previous findings. Ecology and evolution have yet to see the results of such large-scale initiatives. Some large-scale collaborative initiatives have been proposed in ecology and evolution[5], such as ManyPrimates and ManyBirds, but none has yet been implemented. The scope of these projects is specific in terms of their taxonomic coverage and the questions proposed are more about assessing the generalizability than the replicability of previous research results. The lack of replication initiatives may be due to insufficient incentives and field-specific challenges related to the complexities and high costs of replicating studies involving rare species and unique ecosystems[2].

Yet, low replicability is probably pervasive in ecology and evolution. A registered report highlighted widespread publication bias (for example, file-drawer problem), low power (15%) and high inflation of effect (fourfold) across 87 ecological and evolutionary meta-analyses published from 2010 to 2019[2]. Subsequent analyses of 350 recent studies (2018 to 2020) echoed these concerns, revealing prevalent exaggerated effect sizes and publication bias[1,6]. Therefore, indirect evidence has accumulated for low replicability in ecology and evolution, although direct evidence is lacking. Fortunately, recent methodological developments allow us to estimate replicability across the field[7–10].

Here, we use the expansive meta-analysis literature available[2,11] to conduct a large-scale in silico replication project in ecology and evolution. The coverage of this dataset is comprehensive, obtained previously through a systematic search of meta-analyses indexed in Web of Science categories relevant to ecology and evolution, encompassing 88,218 effects (Supplementary Fig. 1) from 12,927 primary studies[12] across a diverse array of research topics within ecology and evolution[2,11]. We summarize our data in terms of the 'true' effect ES, the effect-size estimate $\overline{ES}$ and its standard error SE. Being careful to take the statistical dependence of multiple observations within the same study into account, we obtain the marginal distribution of the $z$ statistics $z = \overline{ES}/SE$ using a Gaussian mixture model. Next, we use a statistical technique called 'deconvolution' to estimate the marginal density of the signal-to-noise ratio SNR = $ES/SE$ (more details in Methods and the reproducible R code at https://yefeng0920.github.io/replication_EcoEvo_git/). As with earlier work[3,4], replicability here is defined as finding a statistically significant effect size in the same direction in an exact replication study (in silico replication). Since the true effects are unobservable[7,9], being able to estimate replicability is remarkable.

The estimated marginal density of the $z$ statistics and SNR is shown in Fig. 1, Extended Data Figs. 1 and 2, Extended Data Table 1 and Supplementary Fig. 2. We constructed the joint distribution between $z$ statistics and SNR (Extended Data Table 1) and the replicability profile corresponding to the observed $z$ statistics (Fig. 1 and Supplementary

[1]Evolution & Ecology Research Centre and School of Biological, Earth and Environmental Sciences, University of New South Wales, Sydney, New South Wales, Australia. [2]Department of Biomedical Data Sciences, Leiden University Medical Center, Leiden, the Netherlands. [3]Department of Statistics and Data Science Institute, University of Chicago, Chicago, IL, USA. [4]Department of Biological Sciences, University of Alberta, Edmonton, Alberta, Canada. [5]These authors contributed equally: Yefeng Yang, Erik van Zwet. ✉e-mail: yefeng.yang1@unsw.edu.au; s.nakagawa@unsw.edu.au

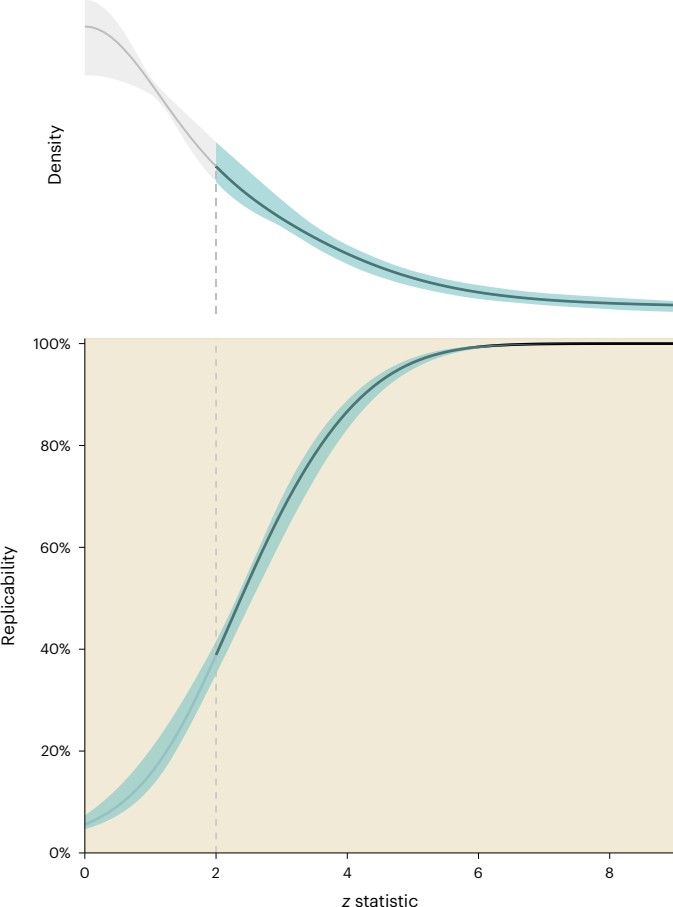

**Fig. 1 | The estimated successful replication probability of 88,218 ecological and evolutionary effects.** A large-scale in silico replication of ecological and evolutionary studies was conducted using the cutting-edge statistical methodologies. The point estimates of replicability and 95% CIs were derived using the 'deconvolution' statistical technique[7,10] and the $F$-localization method[8] (Methods), respectively. Top, the estimated marginal density of the absolute $z$ statistics. Bottom, the probability of successful replication given the observed $z$ statistic of the original study. The line within the shaded area represents the point estimate of successful replication probability. The shaded area represents the corresponding 95% CI. Extended Data Figs. 1 and 4 and Extended Data Table 5 report the replication probability results for the second dataset containing 17,748 ecological and evolutionary effect sizes. See Extended Data Fig. 2 and Extended Data Tables 1–3 for results for different types of effect size metrics.

Fig. 3). As there is a one-to-one correspondence between the $z$ statistic and the two-sided $P$ value under the standard normal distribution, the replicability profile corresponding to the observed $P$ values can also be constructed. We further facilitated the interpretation by categorizing in terms of an informal notion of the strength of statistical evidence against a null hypothesis $H_0$ (debates in refs. 13–17). Finally, we show how replicability increases the larger the sample size of a replication study, relative to the original study (Fig. 2b). All our results are accompanied by 95% confidence intervals (CIs) obtained using the Dvoretzky–Kiefer–Wolfowitz $F$-localization approach (Methods; Supplementary Information)[8].

We found that a study at a significance level ranging from 0.05 to 0.01, which is equivalent to a $z$ statistic between 1.96 and 2.58, had an approximate successful replication probability of 38% (95% CI = [34%–41%]) to 56% (95% CI = [51%–58%]; Fig. 2a and Supplementary Fig. 4), in the absence of selective reporting. This implies that if we randomly select an ecological and evolutionary study with 'moderate' statistical evidence against a null hypothesis $H_0$ an exact replication

study has <50% probability of being successful. Such a replication study would need a sevenfold increase in sample size to achieve a probability of successful replication of 75% (95% CI = [69%–83%]; Fig. 2b). The estimates of replicability corresponding to different types of effect-size measures were also consistent (Extended Data Figs. 2 and 3 and Extended Data Tables 1–4 give standardized mean difference, log response ratio and Fisher's $r$-to-$Zr$; for the detail of each subset dataset see Methods). We successfully repeated the above results in an second dataset containing 17,748 ecological and evolutionary effect sizes from 3,807 meta-analyses (Extended Data Figs. 1 and 4 and Extended Data Table 5; for the detail of this dataset, see Methods), which were collected to maximize the coverage of different topics in ecology and evolution[2] and found very similar results.

Studies with 'strong' statistical evidence against a null hypothesis $H_0$ ($P = 0.001$) showed a replicability of 75% (95% CI = [69%–76%]; Fig. 2, Extended Data Fig. 3 and Supplementary Fig. 4) but would still require at least a twofold increase in sample size to ensure a replicability of around 90% (95% CI = [87%–92%]; Fig. 2, Extended Data Fig. 4 and Supplementary Fig. 5). Only studies with 'very strong' statistical evidence against a null hypothesis $H_0$ ($P = 0.0001$) could achieve replicability as high as 85% (95% CI = [81%–87%]). Among 66,958 statistically significant effects, the average replicability was 77%, assuming no selective reporting exists, which is unlikely (see below)[2,6]. An earlier survey on replication studies conducted in ecology and evolution found only 11 replication studies, with four claiming successful replication[18]. Large-scale replication projects across different disciplines have revealed that around half the effects with $P < 0.05$ could be successfully replicated[4,19]. As in many areas of research, most ecological and evolutionary studies were underpowered (Extended Data Fig. 5 and Supplementary Fig. 4). The immediate consequence of these findings is that statistical significance alone does not provide a guarantee of successful replication, whereas an unsuccessful replication does not mean that the original study was a fluke.

Our study has two important caveats. First, all replicabilities were estimated assuming the absence of publication bias. Because some evidence for publication bias exists[2,6], the estimated replicability here should be interpreted as the upper bound of the true replicability. Second, the estimated replicability assumes that the replication study is an 'idealized exact replication' of the original study (that is, no heterogeneity). Therefore, while the in silico replication approach used in our study provides valuable insights into replicability, we call for a true large-scale replication project, which could eliminate the impact of publication bias and take heterogeneity into account.

The actions for improving replicability have been discussed elsewhere[2]. Replicability can be increased by using larger sample sizes[2] but this costs time and resources[20] and might be impossible when experiments involve rare species and remote and unique ecosystems. At the community level, we encourage coordinated distributed experiments, big-team science and adversarial collaborations[2,20,21]. All well-conducted studies that are supported by sound theory should be published regardless of their statistical significance[20].

At present, single studies often do not have sufficient power to provide definitive answers in ecology and evolution. Therefore, we advocate for emphasizing the experimental design rather than solely focusing on statistical significance[20,21]. Well-designed studies, even with small sample sizes, are not necessarily problematic, if all results (for example, effect-size estimates and CIs), including positive and negative ones, are published to mitigate the file-drawer problem. Meta-analyses can aggregate evidence from those small studies to increase power (and thus replicability). We call for more open science campaigns to eliminate questionable research practices[21], including embracing transparent reporting, (pre)registrations, registered reports, data and code archiving and multiverse analysis[1,2,6]. We encourage ecologists and evolutionary biologists to use the computational methods we

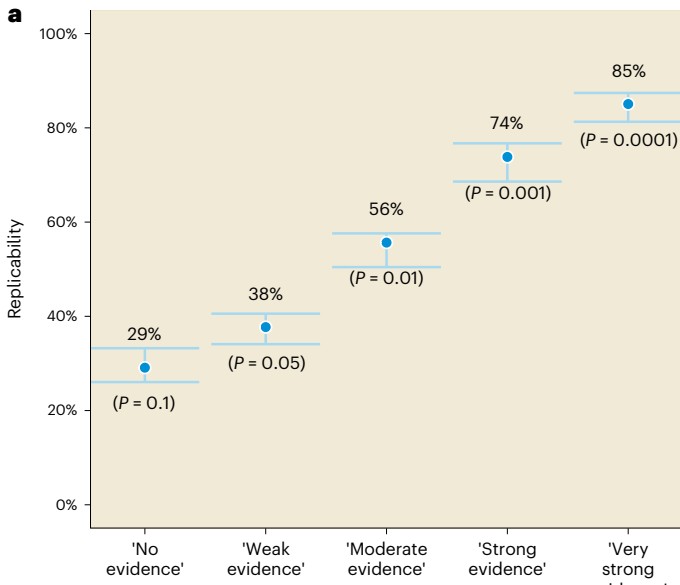

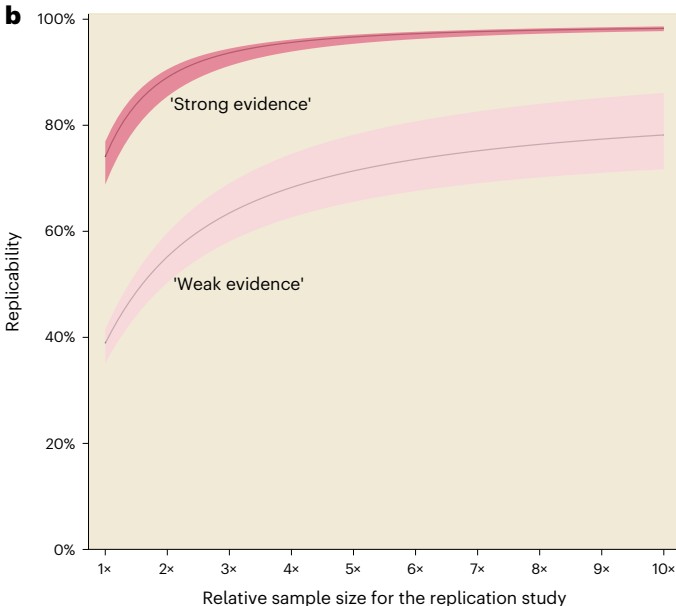

**Fig. 2 | The impact of strength of statistical evidence against a null hypothesis $H_0$ and sample size on the estimated successful replication probability of 88,218 ecological and evolutionary effects. a**, The quantitative relationship between the replication probability and tentative evidence strength benchmarks. **b**, The quantitative relationship between the replication probability and the relative sample size of the replication study compared to the sample size of the original study. We distinguish cases where the original study presented 'weak' and 'strong' statistical evidence. Note the controversy over the informal notion of interpreting $P$ values as measures of the strength of statistical evidence against a null hypothesis $H_0$ (refs. [13–17]). In **a**, the dot within the error band represents the point estimate of the successful replication probability and error band represents the 95% CI derived from the $F$-localization method[8]. In **b**, the line within the shaded area represents the point estimate of successful replication probability and the shaded area represents the 95% CI. The two-sided $P$ value was converted from the $z$ statistic under the standard normal distribution. For other details refer to Fig. 1.

have developed to launch in silico replication projects in their specific topics, although it cannot substitute a true, collaborative replication project. We have implemented them both in R and the Julia language (Supplementary Information; also https://github.com/Yefeng0920/replication_EcoEvo_git).

## Methods

### Database

The database comprised 466 meta-analytic datasets and 111,327 observations of ecological and evolutionary effects, curated by ref. [11] (refer to ref. [11] for a detailed description of their systematic search). Briefly, they searched the Web of Science for 'meta-analys*' AND 'ecol*' while limiting to categories potentially related to ecology (for example, ecology, evolutionary biology and multidisciplinary sciences). The remaining papers were screened for relevance by examining the titles and abstracts. Then, full text, supplements and appendices were reviewed to confirm that the paper addresses an ecological topic and determine if data on effect sizes and their sampling variances were available. Thus, a carefully curated and wide-ranging collection was obtained, which we believe to be representative of the field (at least the field of ecology).

We eliminated effect-size estimates with zero and missing sampling variances and calculated $z$ statistics using the square root of sampling variances, leading to 106,486 $z$ statistics. The 466 meta-analytic datasets encompassed diverse research topics within ecology and evolutionary biology. Although this diversity ensured a comprehensive representation of the field, it also necessitated interpreting our findings as general trends rather than being specific to certain research domains. Given the known prevalence of publication bias in ecology and evolution, the true replicability is likely to be lower than our estimated replicability. Our dataset presented two potential sources of data dependence. The first type was within-study dependence, stemming from the hierarchical structure in which each study contributed several effect-size estimates. To address this, we used a weighting strategy when computing replication probabilities (see next section on 'Estimating joint distributions') and constructing simultaneous CIs (see section on 'Computing confidence intervals for replication probability'). The second type was between-study dependence, arising from the possibility of different meta-analyses sharing the same study. To mitigate this, we identified and removed duplicated studies. After this, our dataset consisted of 88,218 $z$ statistics derived from 12,927 independent studies[12]. We conducted two sensitivity analyses to examine the robustness of our results (see section on 'Robustness check').

### Estimating joint distributions

For all 88,218 observations in our main database, let ES denote the (unobservable) true effect size and $\overline{ES}$ be the observed effect size. $\overline{ES}$ follows a normal distribution with the mean of true effect ES and known sampling variance $V$ (the square of the standard error SE of $\overline{ES}$):

$$\overline{ES}|ES \sim N(ES, V) \qquad (1)$$

The main effect-size measures in our database, included standardized mean differences (SMD; 45%), log-transformed response ratios (lnRR; 36%) and Fisher's $r$-to-$Zr$ coefficients ($Zr$; 15%). The $z$ statistic is defined as $z = \overline{ES}/SE$. If the absolute value of the $z$ statistic exceeds the nominal significance threshold (1.96) then the observed effect is statistically significantly different from zero (two-sided $P < 0.05$). Finally, we defined the signal-noise ratio (SNR) as the true effect size (signal) relative to the standard error of the estimate (noise)[7], that is SNR = ES/SE.

We used a two-step approach to obtain the joint distribution of $z$ and SNR (Extended Data Figs. 1 and 2). First, we modelled the marginal density of $z$ as a mixture of four components of zero-mean normal densities:

$$f(z) = \sum_{k=1}^{4} w_k \varphi(z/\sigma_k)/\sigma_k \qquad (2)$$

where $\varphi$ denotes the standard normal density function, $w_k$ the probability of the $k$th component ($w_k \geq 0$ and $\sum_{k=1}^{4} w_k = 1$) and $\sigma_k$ represents the standard deviation of the $k$th normal distribution ($\sigma_k > 1$). To account for the dependence between multiple effect estimates from the same

study, we weighted each observation $z_{ij}$ of the $i$th unit inverse-proportionally to the number of observations $n_j$ in the same study. We used the maximum likelihood method to estimate weights $w_k$ and standard deviations $\sigma_k$ (Extended Data Tables 1–5). By equation (1), the distribution of the $z$ statistic is the convolution of the distribution of $SNR_i$ and standard normal distribution[7]

$$z|SNR \sim N(SNR, 1) \tag{3}$$

Consequently, we can derive the marginal density of SNR through 'deconvolution' of the estimated density of $z$ (equation (2)) and the standard normal density. This is done by subtracting 1 from the estimated variance of the normal distribution. Thus, we estimate the density of the SNR as

$$g(SNR) = \sum_{k=1}^{4} w_k \varphi(SNR/\tau_k)/\tau_k \tag{4}$$

where $\tau_k = \sqrt{\sigma_k^2 - 1}$. The estimated marginal densities are summarized in Extended Data Fig. 1 and Extended Data Tables 1–5. Besides the two marginal densities, we also have the conditional density of $z$ given SNR. Therefore, we have estimates of the joint density and thus also of the conditional density of SNR given the density of $z$. We used R (v.4.0.1) to estimate the joint distribution of $z$ and SNR.

## Estimating replication probability

We used the estimated joint distribution of $z$ and SNR based on the 106,486 observed effects to estimate the replication probability of ecological and evolutionary studies. Suppose we have conducted a study and obtained a $z$ statistic. Now consider a (hypothetical) replication study with the same specification as the original study (for example, no heterogeneity). We define replication probability (replicability) as the event where the replication reaches statistical significance in the same direction as the original study at the two-sided level $\alpha = 0.05$.

$$z \times z_{repl} > 0 \text{ AND } |z_{repl}| > 1.96 \tag{5}$$

where $z_{repl}$ is the test statistic of the replication study. As we have the joint distribution of the $z$ statistic and the SNR, we can compute the conditional probability of a successful replication given the $z$ statistic of the original study. Note that our notion of replication probability is closely related to the notion of power. In essence, replication probability is the long-run frequency of exact replication studies (that is, replication and original studies are identical in every aspect) having statistical significance in the correct direction when there is a true effect.

Therefore, we also estimated the statistical power, which is defined as the probability of a study reaching statistical significance (two-sided, level $\alpha = 0.05$) when there is a true effect. Statistical power can be expressed in terms of the SNR.

$$\Phi(-1.96 - SNR) + 1 - \Phi(1.96 - SNR) \tag{6}$$

where $\Phi$ represents the cumulative distribution function of the standard normal distribution. We can easily transform our estimate of the distribution of the SNR into an estimate of the distribution of the power. We note that equation (6) yields the statistical power against the true effect ES and should not be confused with the power against an a priori-defined biologically meaningful effect size (power for study design), nor with the power against the observed effect size $\overline{ES}$ (the 'observed' or 'post hoc' power). The power we present includes the probability of a study reaching statistical significance in the wrong direction.

## Predicting the sample size requirements for replication studies

If the sample size of the replication study is $m$ times larger than that of the original study, then the SNR of the replication study will be larger by a factor square root of $m$. Thus, we can also compute the conditional

probability of a successful replication given the $z$ statistic of the original study when the sample size of the replication study is $m$ times larger (with all other aspects of the study remaining identical). We evaluated the impact of increasing the relative sample size corresponding to the $z$ statistics of 1.96 and 3.29, which are interpreted as weak and strong evidence according to the tentative evidence strength benchmarks[17].

## Computing confidence intervals for replication probability

We used the Dvoretzky–Kiefer–Wolfowitz $F$-localization approach[8] to compute CIs at the 95% level for the replication probabilities. The $F$-localization-based CIs provide simultaneous coverage over all replication probabilities (and so, the band in Fig. 1 is a simultaneous confidence band at level 95%). The $F$-localization approach does not require a fixed number of mixture components (in our implementation, we allowed up to 365 mixture components). Although the original Dvoretzky–Kiefer–Wolfowitz $F$-localization approach[8] assumes independence of the $z$ statistic across different studies, we extended it to handle dependent data in our dataset, given that the $z$ statistic from the same study may be arbitrarily correlated in our dataset.

CIs can be computed as follows (for technical details see Extended Data). First, for the marginal distribution of the SNR, we posited the class of all zero-centred Gaussian scale mixtures with scale parameter $\sigma \in \{10^{-5}, 10^{-5} \cdot 1.05, 10^{-5} \cdot 1.05^2, \ldots, 516.3\}$. In other words, the distribution in this class consists of up to 365 mixture components whose standard deviation $\sigma$ takes on values on a logarithmically spaced grid from $10^{-5}$ to 516.3. As mentioned above, our main assumption is that the $z$ statistic is (approximately) normally distributed with mean SNR and variance one. For the $F$-localization approach, we further relaxed the above. We modelled the absolute value of the $z$ statistic ($|z|$), which follows a folded normal distribution and censored the $z$ statistic with absolute values <2.3 and >10. This censorship ensures robustness to mild $P$-hacking close to the nominal significance level cutoff ($|z| \geq 1.96$) to model misspecification near the origin and at outliers for large values of $|z|$. It is important to note that our censoring does not necessarily provide robustness guarantees against selection bias. The mathematical details of the Dvoretzky–Kiefer–Wolfowitz $F$-localization approach for dependent $z$ statistics can be found in the Supplementary Information. We used Julia (v.1.10.0) to construct CIs for replication probability.

## Robustness check

We conducted two sets of sensitivity analyses to examine the robustness of our findings. First, we reclassified the original effect-size measures into three categories: mean difference (including SMD and lnRR; 81%), correlation (Fisher's $r$-to-$Zr$; 15%) and uncommon effect-size measures (for example, regression slope, odds ratio; 4%). Subsequently, we computed replicability estimates only for studies using mean difference and correlation as effect-size measures, respectively, excluding studies with uncommon effect-size measures because of their low occurrence. Second, we replicated our main analysis using an independent ecological and evolutionary meta-analytic dataset with 17,748 ecological and evolutionary effect-size estimates[2]. This second dataset was initially used to examine the degree of publication bias and effect-size inflation in ecology and evolution. In brief, the initial database creation process involved compiling a list of journals in the categories of 'ecology' and/or 'evolutionary biology' using the ISI InCites Journal Citation Reports. Subsequently, a search was conducted in Scopus using specific strings related to meta-analysis. The search was limited to articles published from January 2010 to 25 March 2019. Results were filtered to identify the 31 journals with the highest frequency of publishing meta-analyses. A random sample of studies from each of these journals resulted in a total of 297 papers. Following screening, the database ultimately comprised a representative sample of 102 ecological or evolutionary meta-analyses; therefore, these meta-analyses are likely to be representative of the field of both ecology and evolution. We manually compared the titles of meta-analysis

papers from this independent dataset with those from the main dataset to remove the duplicated studies, making sure the two datasets were independent. The replicated results were consistent with those from our main analysis (Extended Data Figs. 3–5).

### Reporting summary

Further information on research design is available in the Nature Portfolio Reporting Summary linked to this article.

## Data availability

The data to reproduce the results of this study are available at https://github.com/Yefeng0920/replication_EcoEvo_git). The data are also available via Zenodo at https://doi.org/10.5281/zenodo.12748092 (ref. 12).

## Code availability

The code for reproducing the results of this study is available at https://github.com/Yefeng0920/replication_EcoEvo_git. The code is also available via Zenodo at https://doi.org/10.5281/zenodo.12748092 (ref. 12). We provide code implemented in R (v.4.0.1) and Julia (v.1.10.0). The reproducible R code in an interactive format (code chunks paired with results) also can be found in the Supplementary Information and at https://yefeng0920.github.io/replication_EcoEvo_git/.

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

## Acknowledgements

We thank T. H. Parker, A. Sánchez-Tójar, R. E. O'Dea and R. Spake for their efforts on the earlier version of the draft. Y.Y. was funded by the National Natural Science Foundation of China (no. 32102597). Y.Y. and S.N. were funded by the Australian Research Council Discovery grant nos. DP210100812 and DP230101248.

## Author contributions

Y.Y. collected the data. All authors conceived the study, analysed the data and wrote and edited the manuscript.

## Funding

## Competing interests

The authors declare no competing interests.

## Additional information

**Extended data** is available for this paper at https://doi.org/10.1038/s41559-024-02530-5.

**Correspondence and requests for materials** should be addressed to Yefeng Yang or Shinichi Nakagawa.

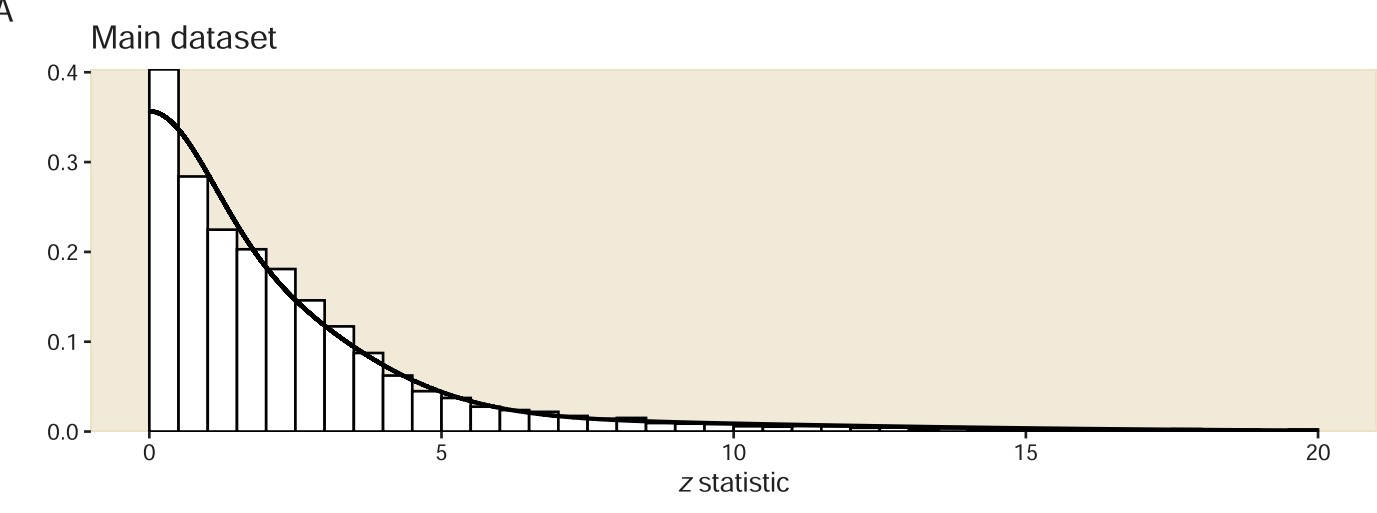

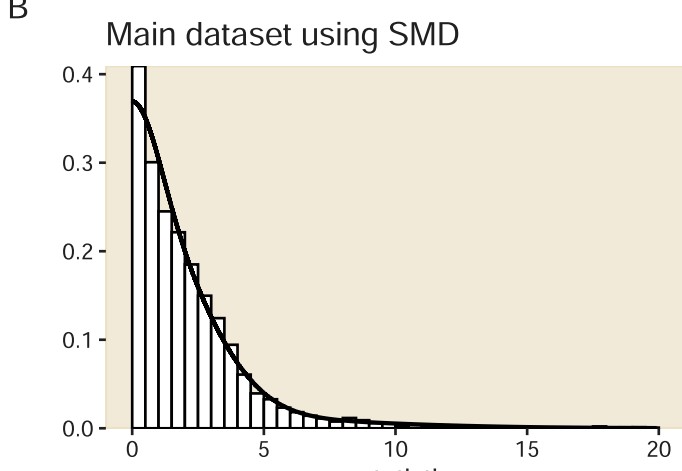

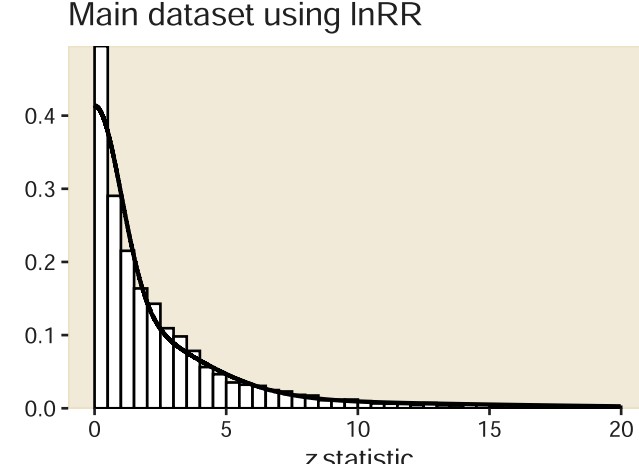

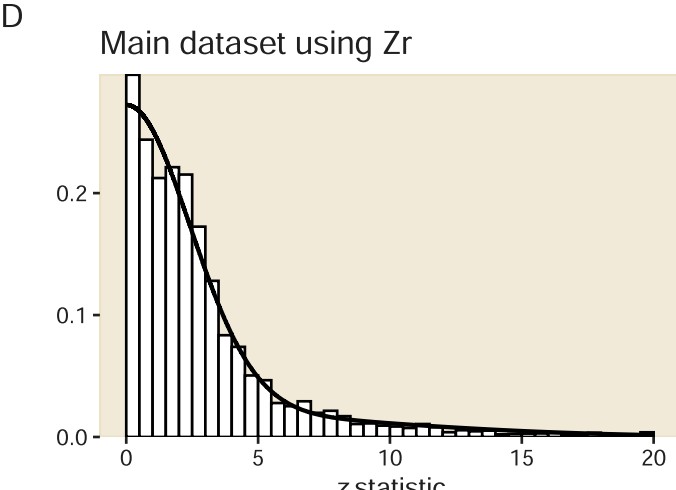

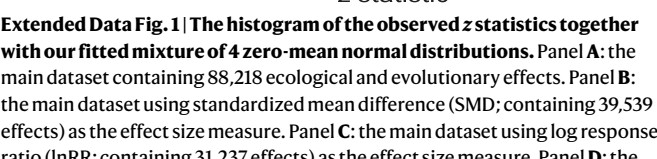

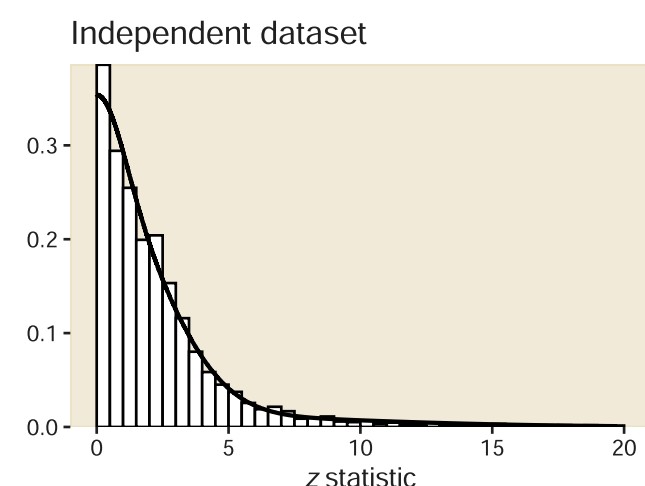

**Extended Data Fig. 1 | The histogram of the observed z statistics together with our fitted mixture of 4 zero-mean normal distributions.** Panel **A**: the main dataset containing 88,218 ecological and evolutionary effects. Panel **B**: the main dataset using standardized mean difference (SMD; containing 39,539 effects) as the effect size measure. Panel **C**: the main dataset using log response ratio (lnRR; containing 31,237 effects) as the effect size measure. Panel **D**: the main dataset using Fisher's r-to-Zr (Zr; containing 13,965 effects) as the effect size measure. Panel **E**: a second dataset containing 17,748 ecological and evolutionary effects. The specifications of the fitted mixture model can be found in Methods. The detailed parameter estimates for the fitted mixture model can be found in Extended Data Tables 1 to 5.

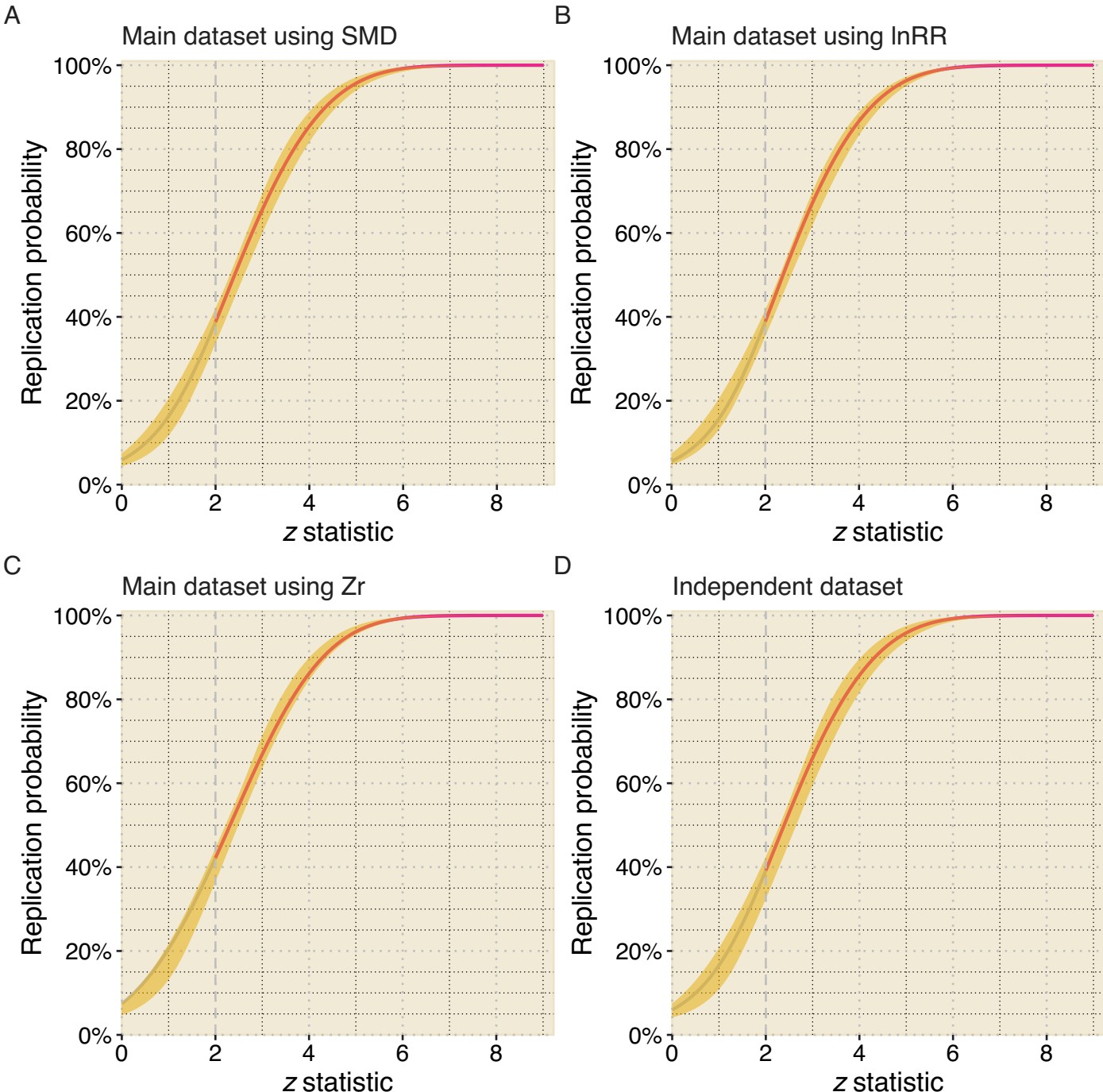

**Extended Data Fig. 2 | The estimated probability of successful replication given the observed z statistic of the original study.** The shaded area represents the 95% confidence interval (see Methods). Panel **A**: the replicability of the main dataset using SMD (containing 39,539 effects) as the effect size measure. Panel **B**: the replicability of the main dataset using lnRR (containing 31,237 effects) as the effect size measure. Panel **C**: the replicability of the main dataset using *Zr* (containing 13,965 effects) as the effect size measure. Panel **D**: the replicability of a second dataset containing 17,748 ecological and evolutionary effects. The line within the shaded area represents the point estimate of successful replication probability, and the shaded area represents the 95% confidence interval. For other details refer to Fig. 1, and Extended Data Fig. 1.

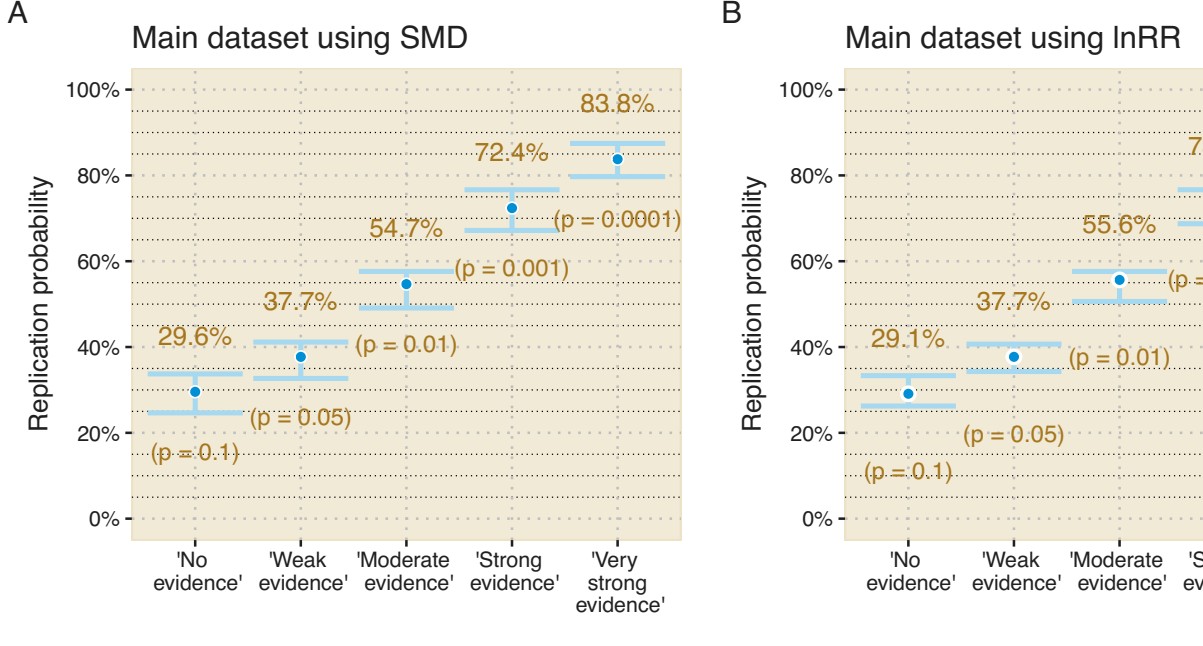

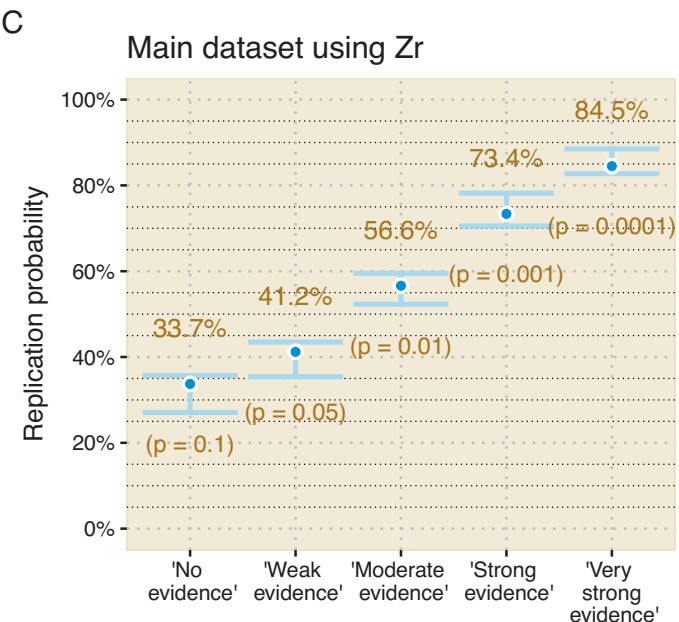

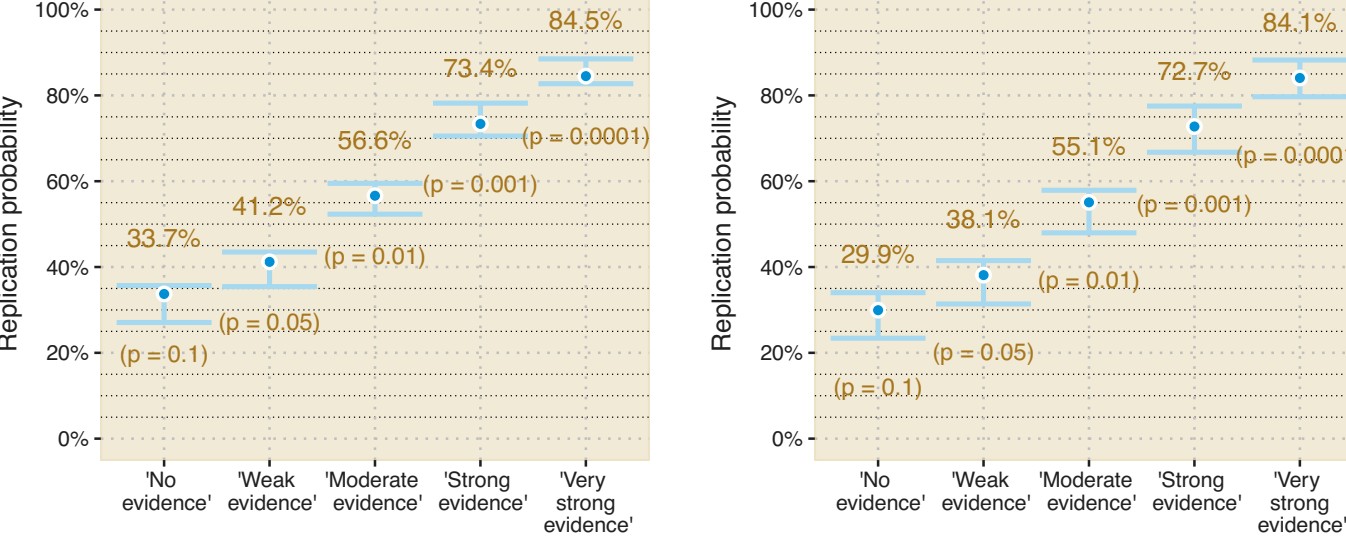

**Extended Data Fig. 3 | The quantitative relationship between the successful replication probability and tentative evidence strength benchmarks.**
Panel **A**: the relationship derived from the main dataset using SMD (containing 39,539 effects) as the effect size measure. Panel **B**: the relationship derived from the main dataset using lnRR (containing 31,237 effects) as the effect size measure. Panel **C**: the relationship derived from the main dataset using *Zr* (containing 13,965 effects) as the effect size measure. Panel **D**: the relationship derived

from a second dataset containing 17,748 ecological and evolutionary effects. the dot within the error band represents the point estimate of the successful replication probability, and error band represents the 95% confidence interval (see Methods). The two-sided p-value was converted from the z statistic under the standard normal distribution. For other details refer to Fig. 1, and Extended Data Figs. 1 and 2.

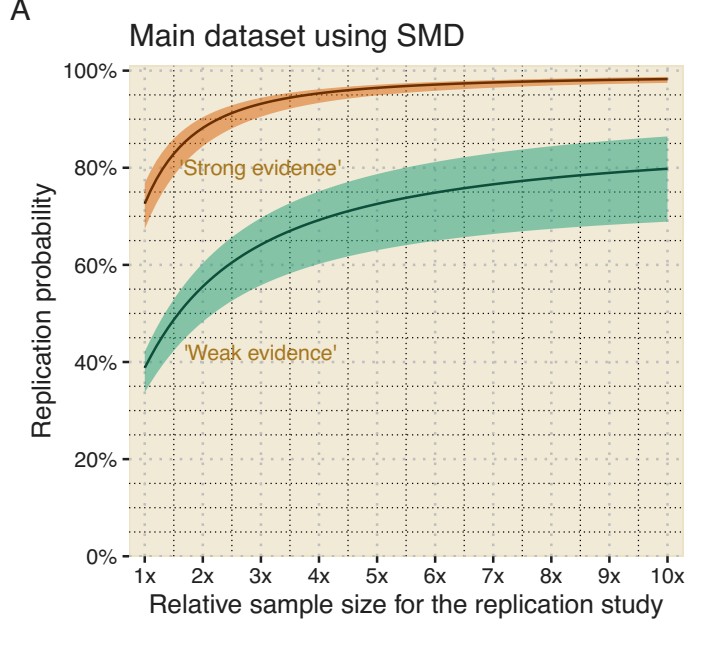

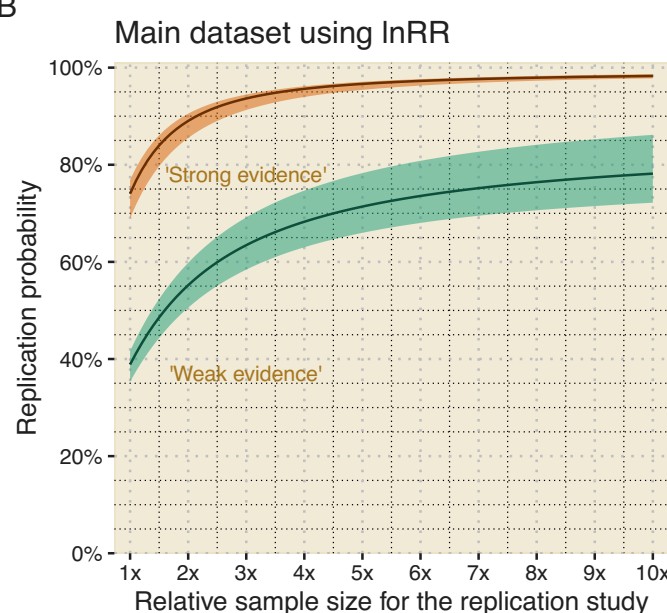

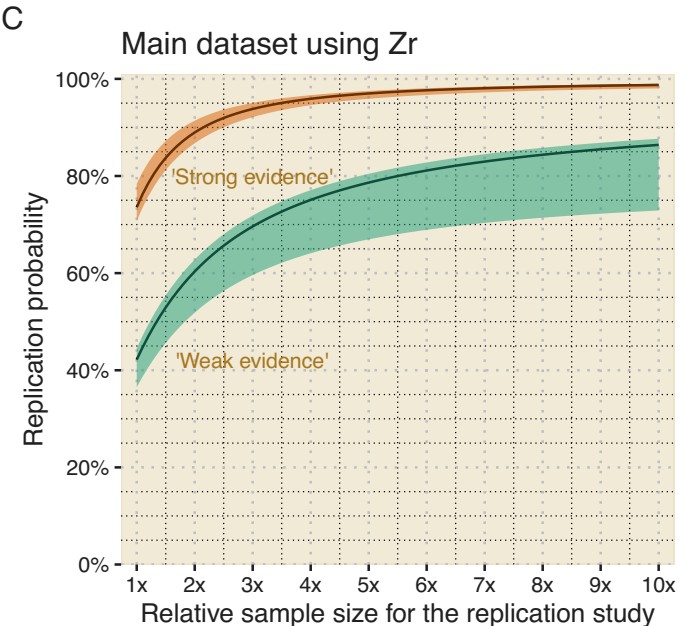

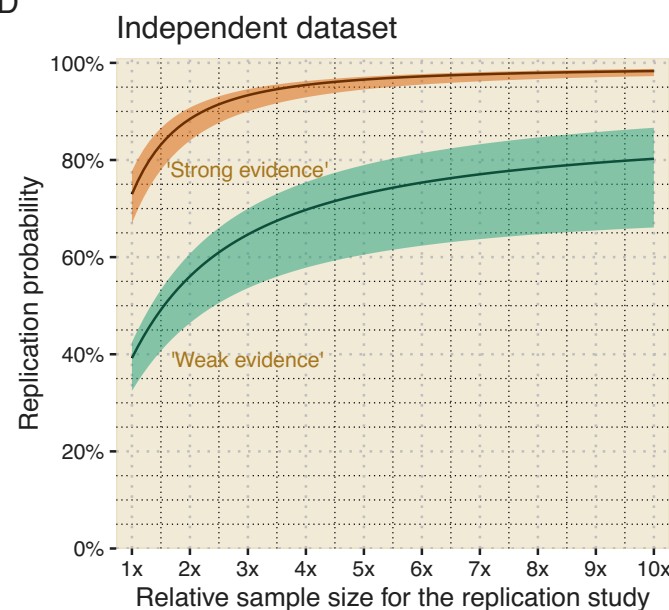

**Extended Data Fig. 4 | The quantitative relationship between the successful replication probability and the relative sample size of the replication study compared to the sample size of the original study.** Panel **A**: the relationship derived from the main dataset using SMD (containing 39,539 effects) as the effect size measure. Panel **B**: the relationship derived from the main dataset using lnRR (containing 31,237 effects) as the effect size measure. Panel **C**: the relationship derived from the main dataset using *Zr* (containing 13,965 effects) as the effect size measure. Panel **D**: the relationship derived from a second dataset containing 17,748 ecological and evolutionary effects. the line within the shaded area represents the point estimate of successful replication probability, and the shaded area represents the 95% confidence interval (see Methods). For other details refer to Fig. 1, and Extended Data Figs. 1 and 2.

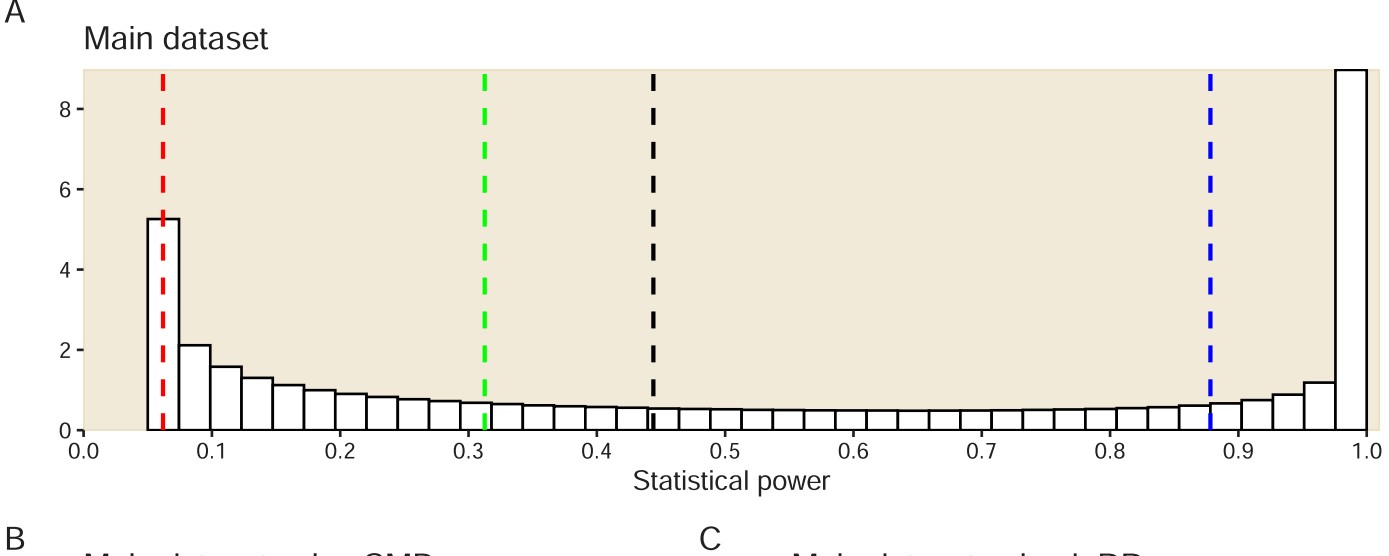

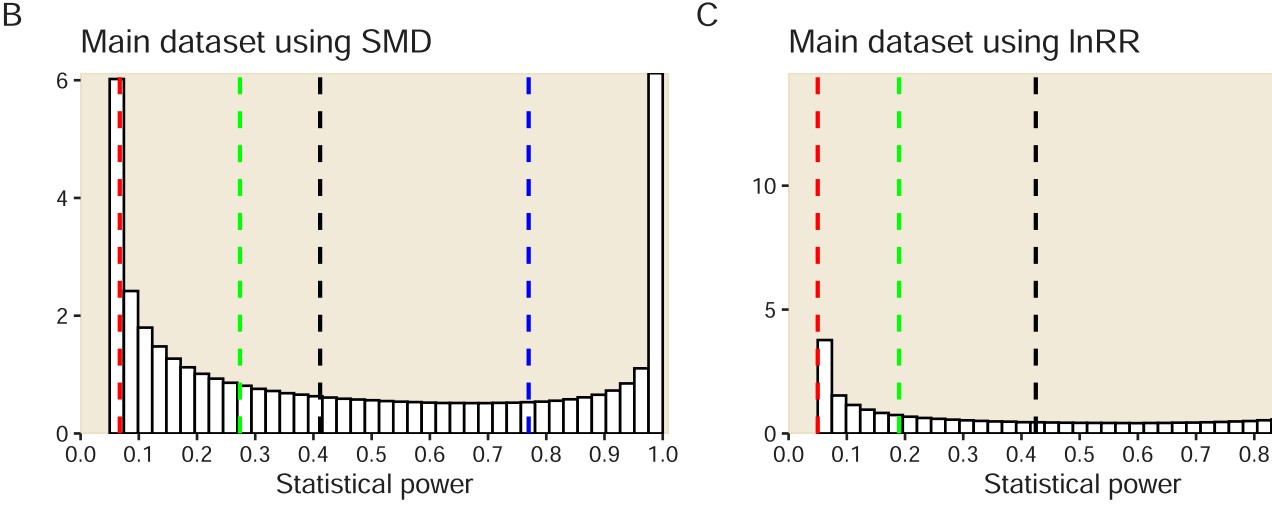

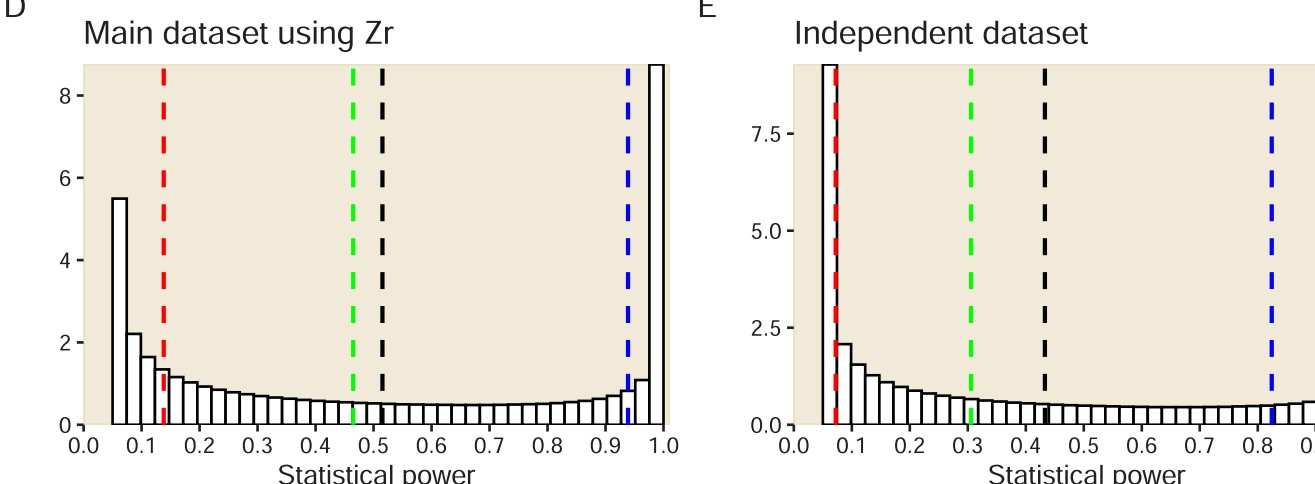

**Extended Data Fig. 5 | The estimated distribution of the statistical power against the true effect.** Panel **A**: the statistical power of the main dataset (containing 88,218 effects). Panel **B**: the statistical power of the main dataset using SMD (containing 39,539 effects) as the effect size measure. Panel **C**: the statistical power of the main dataset using lnRR (containing 31,237 effects) as the effect size measure. Panel **D**: the statistical power of the main dataset using

Zr (containing 13,965 effects) as the effect size measure. Panel **E**: the statistical power of a second dataset containing 17,748 ecological and evolutionary effects. The red, green, black, and blue dashed lines represent the first quartile, the second quartile (median), the mean value, and the third quartile. For other details refer to Fig. 1, and Extended Data Figs. 1 and 2.

**Extended Data Table 1 | The estimated *4*-component zero-mean mixture distributions of the *z* statistic and signal-noise ratio (*SNR*) of the main dataset containing 88,218 ecological and evolutionary effects**

| Parameters | Component of the mixture distribution | | | |
|---|---|---|---|---|
| | 1st | 2nd | 3rd | 4th |
| $w$ | 0.21 | 0.17 | 0.43 | 0.18 |
| $m$ | 0.00 | 0.00 | 0.00 | 0.00 |
| $\sigma$ | 2.52 | 1.00 | 2.56 | 8.47 |
| $\tau$ | 2.32 | 0.002 | 0.36 | 8.41 |

The *4*-component mixture was fitted using Eqs. 2 and 4 in Methods. *w* denotes the weight or proportion of the *k*-th (*k*=1,2,3,4) normal distribution. *m* denotes the mean of the *k*-th normal distribution. σ denotes the standard deviation corresponding to the *k*-th normal distribution of the *z* statistics. τ denotes the standard deviation corresponding to the *k*-th normal distribution of the *SNRs*. The z statistic is the sum of the *SNR* and an independent, Gaussian error term.

**Extended Data Table 2 | The estimated 4-component zero-mean mixture distributions of the z statistic and signal-noise ratio (SNR) of the main dataset using standardized mean difference (SMD) as an effect size measure**

| Parameters | Component of the mixture distribution | | | |
|---|---|---|---|---|
| | 1st | 2nd | 3rd | 4th |
| $w$ | 0.57 | 0.15 | 0.14 | 0.14 |
| $m$ | 0.00 | 0.00 | 0.00 | 0.00 |
| $\sigma$ | 2.40 | 2.42 | 1.00 | 6.70 |
| $\tau$ | 2.19 | 2.21 | 0.0002 | 6.63 |

For other details refer to Extended Data Table 1.

**Extended Data Table 3 | The estimated *4*-component zero-mean mixture distributions of the *z* statistic and signal-noise ratio (*SNR*) of the main dataset using log response ratio (lnRR) as an effect size measure**

| Parameters | Component of the mixture distribution | | | |
|:---:|:---:|:---:|:---:|:---:|
| | 1st | 2nd | 3rd | 4th |
| $w$ | 0.46 | 0.26 | 0.11 | 0.17 |
| $m$ | 0.00 | 0.00 | 0.00 | 0.00 |
| $\sigma$ | 3.31 | 1.00 | 1.00 | 11.60 |
| $\tau$ | 3.16 | 0.002 | 0.002 | 11.56 |

For other details refer to Extended Data Table 1.

**Extended Data Table 4 | The estimated *4*-component zero-mean mixture distributions of the *z* statistic and signal-noise ratio (*SNR*) of the main dataset using Fisher's r-to-*Zr* (*Zr*) as an effect size measure**

| Parameters | Component of the mixture distribution | | | |
|:---:|:---:|:---:|:---:|:---:|
| | 1st | 2nd | 3rd | 4th |
| $w$ | 0.42 | 0.35 | 0.00 | 0.24 |
| $m$ | 0.00 | 0.00 | 0.00 | 0.00 |
| $\sigma$ | 2.31 | 2.60 | 2.68 | 8.70 |
| $\tau$ | 2.08 | 2.40 | 2.49 | 8.64 |

For other details refer to Extended Data Table 1.

**Extended Data Table 5 | The estimated *4*-component zero-mean mixture distributions of the *z* statistic and signal-noise ratio (*SNR*) of a second dataset containing 17,748 ecological and evolutionary effects**

| Parameters | Component of the mixture distribution | | | |
|---|---|---|---|---|
| | 1st | 2nd | 3rd | 4th |
| $w$ | 0.00 | 0.13 | 0.72 | 0.15 |
| $m$ | 0.00 | 0.00 | 0.00 | 0.00 |
| $\sigma$ | 6.11 | 1.00 | 2.45 | 8.32 |
| $\tau$ | 6.03 | 0.02 | 2.23 | 8.26 |

For other details refer to Extended Data Table 1.

**nature** portfolio

# Reporting Summary

## Statistics

For all statistical analyses, confirm that the following items are present in the figure legend, table legend, main text, or Methods section.

| n/a | Confirmed | |
|---|---|---|
| ☐ | ☒ | The exact sample size (*n*) for each experimental group/condition, given as a discrete number and unit of measurement |
| ☒ | ☐ | A statement on whether measurements were taken from distinct samples or whether the same sample was measured repeatedly |
| ☐ | ☒ | The statistical test(s) used AND whether they are one- or two-sided<br>*Only common tests should be described solely by name; describe more complex techniques in the Methods section.* |
| ☐ | ☒ | A description of all covariates tested |
| ☐ | ☒ | A description of any assumptions or corrections, such as tests of normality and adjustment for multiple comparisons |
| ☐ | ☒ | A full description of the statistical parameters including central tendency (e.g. means) or other basic estimates (e.g. regression coefficient) AND variation (e.g. standard deviation) or associated estimates of uncertainty (e.g. confidence intervals) |
| ☐ | ☒ | For null hypothesis testing, the test statistic (e.g. *F*, *t*, *r*) with confidence intervals, effect sizes, degrees of freedom and *P* value noted<br>*Give P values as exact values whenever suitable.* |
| ☒ | ☐ | For Bayesian analysis, information on the choice of priors and Markov chain Monte Carlo settings |
| ☒ | ☐ | For hierarchical and complex designs, identification of the appropriate level for tests and full reporting of outcomes |
| ☐ | ☒ | Estimates of effect sizes (e.g. Cohen's *d*, Pearson's *r*), indicating how they were calculated |

*Our web collection on statistics for biologists contains articles on many of the points above.*

## Software and code

Policy information about availability of computer code

| | |
|---|---|
| Data collection | All data are extracted from meta-analysis articles indexed in the WoS for the search term "meta-analys*" AND "ecol*" while limiting to categories potentially related to ecology (e.g., Ecology, Evolutionary Biology, Multidisciplinary Sciences). No software was used for data collection. |
| Data analysis | All data processes and analyses are done using R programming language (4.0.1) and Julia programming Language (1.10.0). The code for reproducing the results of this study are available at the GitHub repository (https://github.com/Yefeng0920/replication_EcoEvo_git). The code also can be found at Zenodo repository 12 (https://doi.org/10.5281/zenodo.12748092). We provide both the R code and Julia code. The reproducible R code in an interactive format (code chunks paired with results) also can be found at the website https://yefeng0920.github.io/replication_EcoEvo_git/). |

For manuscripts utilizing custom algorithms or software that are central to the research but not yet described in published literature, software must be made available to editors and reviewers. We strongly encourage code deposition in a community repository (e.g. GitHub). See the Nature Portfolio guidelines for submitting code & software for further information.

# Data

Policy information about availability of data

All manuscripts must include a data availability statement. This statement should provide the following information, where applicable:

- Accession codes, unique identifiers, or web links for publicly available datasets
- A description of any restrictions on data availability
- For clinical datasets or third party data, please ensure that the statement adheres to our policy

> The data to reproduce the results of this study are available at the GitHub repository (https://github.com/Yefeng0920/replication_EcoEvo_git). The data also can be found at Zenodo repository (https://doi.org/10.5281/zenodo.12748092).

# Research involving human participants, their data, or biological material

Policy information about studies with human participants or human data. See also policy information about sex, gender (identity/presentation), and sexual orientation and race, ethnicity and racism.

| | |
|---|---|
| Reporting on sex and gender | NA |
| Reporting on race, ethnicity, or other socially relevant groupings | NA |
| Population characteristics | NA |
| Recruitment | NA |
| Ethics oversight | NA |

Note that full information on the approval of the study protocol must also be provided in the manuscript.

# Field-specific reporting

Please select the one below that is the best fit for your research. If you are not sure, read the appropriate sections before making your selection.

☐ Life sciences          ☐ Behavioural & social sciences          ☒ Ecological, evolutionary & environmental sciences

For a reference copy of the document with all sections, see nature.com/documents/nr-reporting-summary-flat.pdf

# Ecological, evolutionary & environmental sciences study design

All studies must disclose on these points even when the disclosure is negative.

| | |
|---|---|
| Study description | We summarize our data in terms of the "true" effect ES, the effect size estimate $(\hat{ES})$, and its standard error SE. While being careful to take the statistical dependence of multiple observations within the same study into account, we obtain the marginal distribution of the z statistics $z=(\hat{ES})/SE$ using a Gaussian mixture model. Next, we employ a statistical technique called "deconvolution" to estimate the marginal density of the signal-noise ratio $SNR=ES/SE$. As with earlier work, replicability here is defined as finding a statistically significant effect size in the same direction in an exact replication study (in silico replication). Since the true effects are unobservable, being able to estimate replicability is rather remarkable. |
| Research sample | All data are extracted from meta-analysis articles indexed in the WoS for the search term "meta-analys*" AND "ecol*" while limiting to categories potentially related to ecology (e.g., Ecology, Evolutionary Biology, Multidisciplinary Sciences). |
| Sampling strategy | The coverage of this dataset is comprehensive, obtained previously through a systematic search of meta-analyses indexed in Web of Science categories relevant to ecology and evolution, encompassing 88,218 effects from 12,927 primary studies across a diverse array of research topics within ecology and evolution. |
| Data collection | The database comprised 466 meta-analytic datasets of ecological and evolutionary effects, curated by The database comprised 466 meta-analytic datasets and 111,327 observations of ecological and evolutionary effects, curated by Costello and Fox (ref 10). |
| Timing and spatial scale | Meta-analysis papers published before 2022. |
| Data exclusions | full text, supplements, and appendices were reviewed to confirm that the paper addresses an ecological topic and determine if data on effect sizes and their sampling variances were available. |
| Reproducibility | The authors affirms that the manuscript is an honest, accurate, and transparent account of the study being reported, and no important aspects of the study have been omitted. The data and code to reproduce the results of this study are available at the GitHub repository (https://github.com/Yefeng0920/replication_EcoEvo_git). We provide both the R code and Julia code. The |

reproducible R code in an interactive format (HTML) also can be found at the Supplementary material and https://yefeng0920.github.io/replication_EcoEvo_git/). The code also can be found at Zenodo repository (https://doi.org/10.5281/zenodo.12748092).

| | |
|---|---|
| Randomization | NA |
| Blinding | NA |

Did the study involve field work?  ☐ Yes  ☒ No

# Reporting for specific materials, systems and methods

We require information from authors about some types of materials, experimental systems and methods used in many studies. Here, indicate whether each material, system or method listed is relevant to your study. If you are not sure if a list item applies to your research, read the appropriate section before selecting a response.

## Materials & experimental systems

| n/a | Involved in the study |
|---|---|
| ☒ ☐ | Antibodies |
| ☒ ☐ | Eukaryotic cell lines |
| ☒ ☐ | Palaeontology and archaeology |
| ☒ ☐ | Animals and other organisms |
| ☒ ☐ | Clinical data |
| ☒ ☐ | Dual use research of concern |
| ☒ ☐ | Plants |

## Methods

| n/a | Involved in the study |
|---|---|
| ☒ ☐ | ChIP-seq |
| ☒ ☐ | Flow cytometry |
| ☒ ☐ | MRI-based neuroimaging |

## Plants

| | |
|---|---|
| Seed stocks | NA |
| Novel plant genotypes | NA |
| Authentication | NA |

