## [Peer Review File · Nature Ecology & Evolution]

Peer Review Information

Journal: Nature Ecology & Evolution

Manuscript Title: A large-scale in silico replication of ecological and evolutionary studies

Corresponding author name(s): Yefeng Yang, Shinichi Nakagawa

Editorial Notes:

Reviewer Comments & Decisions:

Decision Letter, initial version:

Our ref: NATECOLEVOL-24051397

2nd July 2024

Dear Dr. Yang,

Thank you for your patience as we've prepared the guidelines for final submission of your Nature Ecology & Evolution manuscript, "A large-scale in silico replication of ecological and evolutionary studies" (NATECOLEVOL-24051397). Please carefully follow the step-by-step instructions provided in the attached file, and add a response in each row of the table to indicate the changes that you have made. Please also check and comment on any additional marked-up edits we have proposed within the text. Ensuring that each point is addressed will help to ensure that your revised manuscript can be swiftly handed over to our production team.

****We would like to start working on your revised paper, with all of the requested files and forms, as soon as possible (preferably within two weeks). Please get in contact with us immediately if you anticipate it taking more than two weeks to submit these revised files.****

In recognition of the time and expertise our reviewers provide to Nature Ecology & Evolution's editorial process, we would like to formally acknowledge their contribution to the external peer review of your

2manuscript entitled "A large-scale in silico replication of ecological and evolutionary studies". For those reviewers who give their assent, we will be publishing their names alongside the published article.

Nature Ecology & Evolution offers a Transparent Peer Review option for new original research manuscripts submitted after December 1st, 2019. As part of this initiative, we encourage our authors to support increased transparency into the peer review process by agreeing to have the reviewer comments, author rebuttal letters, and editorial decision letters published as a Supplementary item. When you submit your final files please clearly state in your cover letter whether or not you would like to participate in this initiative. Please note that failure to state your preference will result in delays in accepting your manuscript for publication.

Cover suggestions

We welcome submissions of artwork for consideration for our cover. For more information, please see our guide for cover artwork.

Nature Ecology & Evolution has now transitioned to a unified Rights Collection system which will allow our Author Services team to quickly and easily collect the rights and permissions required to publish your work. Approximately 10 days after your paper is formally accepted, you will receive an email in providing you with a link to complete the grant of rights. If your paper is eligible for Open Access, our Author Services team will also be in touch regarding any additional information that may be required to arrange payment for your article.

Please note that *Nature Ecology & Evolution* is a Transformative Journal (TJ). Authors may publish their research with us through the traditional subscription access route or make their paper immediately open access through payment of an article-processing charge (APC). Authors will not be required to make a final decision about access to their article until it has been accepted. Find out more about Transformative Journals

Authors may need to take specific actions to achieve compliance with funder and institutional open access mandates. If your research is supported by a funder that requires immediate open access (e.g. according to Plan S principles) then you should select the gold OA route, and we will direct you to the compliant route where possible. For authors selecting the subscription publication route, the journal's standard licensing terms will need to be accepted, including <https://www.nature.com/nature-portfolio/editorial-policies/self-archiving-and-license-to-publish>. Those licensing terms will supersede any other terms that the author or any third party may assert apply to any version of the manuscript.

[REDACTED]

[REDACTED]

Reviewer #1:

Remarks to the Author:

In “A large-scale in silico replication of ecological and evolutionary studies” the authors present a novel (to me) approach to estimate the replicability of empirical findings through simulating exact replications of results from a large dataset of extracted summary statistics from published meta-analyses with over 12,000 original studies and over 88,000 observations (after removal of duplicates). The methods provide a credible estimate of typical replicability (and that is in fact a high end estimate given the likely effect that publication bias has on the the studies represented in the literature), which varies from around 29% for studies with p-values around .1, to 38% at the typical .05 level, up to 85% at $p=.0001$. I have no substantive comments to suggest that this paper should not be published as it provides a reasonable characterization that is backed up by empirical estimates that use new, original data in other fields. Given the difficulty in actually conducting large scale, direct replications, this computer simulation using available data fills an important gap. Below are a few modest suggestions for the authors to consider.

3As a matter of personal policy, I sign my reviews,

David Mellor

Director of Policy, Center for Open Science, <https://orcid.org/0000-0002-3125-5888>

Minor comments:

“We facilitated the interpretation by categorizing in terms of an informal notion of the strength of statistical evidence” <https://www.tandfonline.com/doi/full/10.1080/00031305.2016.1154108>

As the authors likely know, it is problematic to assign terms such as “weak” or “strong” evidence based on p-value in the absence of other information about the study, but their effort to be succinct is appropriate because there is no more clear method of conveying this problem. I recommend putting these terms in quotation marks in order to emphasize to the reader that these are indeed informal notions: “weak” evidence would be better than simply stating weak evidence, etc.

“Well-designed studies, even with small sample sizes, may not necessarily be problematic because meta-analyses can effectively increase power (and thus replicability).” I disagree with the first clause of this sentence if these underpowered studies still report p-values when inferences or conclusions from the small study are inappropriately made. I would recommend that the authors amend this recommendation to assert that well designed, but small studies are underpowered, but should still be published in order to be later used in meta-analyses, as long as inferential statistics are not used in them. This is necessary, in my opinion, because those single study, small sample papers are still standalone research outputs and are therefore open to misinterpretation prior to the meta-analysis being conducted (or discovered by a reader who is unaware of the meta-analysis).

Reviewer #2:

Remarks to the Author:

This study attempts to estimate the replicability of studies in ecological and evolutionary studies. The manuscript is well written, and both the manuscript and the abstract appear to be clear and accessible.

While using studies from meta-analyses can be an efficient and time saving approach, it is not possible to rule out that meta-analyses will contain studies that have been selected based on their effect size and not based on their quality. Thus, there is reason to believe that their sample might not be representative for the literature as a whole. This greatly weakens the strength of the evidence provided and creates

4uncertainty about the result.

The authors state that their result is to be viewed as an upper bound of the true replicability due to publication bias. This is probably true but considering that one must expect large amounts of publication bias, then it diminishes the usefulness of the findings of the study, as the true replicability can be quite far off from the estimate.

I would also suggest citing some wider literature related to replicability and signal to noise ratio, as the authors are mainly citing their own papers and some replication projects and not many related papers are cited.

As the authors state, their study cannot replace substitute a true, collaborative replication project, and I am unsure of the real contribution this makes to the literature as it is already well known that the reproducibility is considerably lower than the acceptable rate in most fields (see some of the replication projects the manuscript is citing) and this manuscript provides a highly uncertain estimate of the replicability.

Author Rebuttal to Initial comments

Reviewer #1:

Remarks to the Author:

In “A large-scale in silico replication of ecological and evolutionary studies” the authors present a novel (to me) approach to estimate the replicability of empirical findings through simulating exact replications of results from a large dataset of extracted summary statistics from published meta-analyses with over 12,000 original studies and over 88,000 observations (after removal of duplicates). The methods provide a credible estimate of typical replicability (and that is in fact a high end estimate given the likely effect that publication bias has on the the studies represented in the literature), which varies from around 29% for studies with p-values around .1, to 38% at the typical .05 level, up to 85% at $p=.0001$. I have no substantive comments to suggest that this paper should not be published as it provides a reasonable characterization that is backed up by empirical estimates that use new, original data in other fields. Given the difficulty in actually conducting large scale, direct replications, this computer simulation using available data fills an important gap. Below are a few modest suggestions for the authors to consider.

As a matter of personal policy, I sign my reviews,

5David Mellor

Director of Policy, Center for Open Science, <https://orcid.org/0000-0002-3125-5888>

REPLY 1: We thank Reviewer 1 for recognizing the value of our manuscript and for the precise summary provided.

#-----

Minor comments:

“We facilitated the interpretation by categorizing in terms of an informal notion of the strength of statistical evidence” <https://www.tandfonline.com/doi/full/10.1080/00031305.2016.1154108>

As the authors likely know, it is problematic to assign terms such as “weak” or “strong” evidence based on p-value in the absence of other information about the study, but their effort to be succinct is appropriate because there is no more clear method of conveying this problem. I recommend putting these terms in quotation marks in order to emphasize to the reader that these are indeed informal notions: “weak” evidence would be better than simply stating weak evidence, etc.

REPLY 2: As suggested, we have added quotation marks to terms representing the informal notion of the strength of statistical evidence throughout the revised manuscript (see lines 94, 108, and 112, and the legend of Figure 2).

#-----

“Well-designed studies, even with small sample sizes, may not necessarily be problematic because meta-analyses can effectively increase power (and thus replicability).” I disagree with the first clause of this sentence if these underpowered studies still report p-values when inferences or conclusions from the small study are inappropriately made. I would recommend that the authors amend this recommendation to assert that well designed, but small studies are underpowered, but should still be published in order to be later used in meta-analyses, as long as inferential statistics are not used in them. This is necessary, in my opinion, because those single study, small sample papers are still standalone research outputs and are therefore open to misinterpretation prior to the meta-analysis being conducted (or discovered by a reader who is unaware of the meta-analysis).

REPLY 3: We appreciate Reviewer 1’s insights. The following changes were made to incorporate this comment:

“Well-designed studies, even with small sample sizes, are not necessarily problematic, if all results (e.g., effect size estimates and confidence intervals), including positive and negative ones, are

published to mitigate the file-drawer problem. Meta-analyses can aggregate evidence from those small studies to increase power (and thus replicability).” (lines 145 – 149 in the revised version)

#-----

Reviewer #2:

Remarks to the Author:

This study attempts to estimate the replicability of studies in ecological and evolutionary studies. The manuscript is well written, and both the manuscript and the abstract appear to be clear and accessible.

REPLY 4: We thank Reviewer 2 for the positive comments.

#-----

While using studies from meta-analyses can be an efficient and time saving approach, it is not possible to rule out that meta-analyses will contain studies that have been selected based on their effect size and not based on their quality. Thus, there is reason to believe that their sample might not be representative for the literature as a whole. This greatly weakens the strength of the evidence provided and creates uncertainty about the result.

REPLY 5: The brief data collection process outlined in the subsection Database of Methods clearly indicates that a carefully curated and wide-ranging collection of meta-analyses was obtained, which we believe to be representative of the field (at least the field of ecology).

#-----

The authors state that their result is to be viewed as an upper bound of the true replicability due to publication bias. This is probably true but considering that one must expect large amounts of publication bias, then it diminishes the usefulness of the findings of the study, as the true replicability can be quite far off from the estimate.

REPLY 6: Across literature, there is no evidence to assume large amounts of publication bias in ecology and evolution. Importantly, in our initial manuscript, we have emphasized in multiple places that the replicability was estimated under the absence of selective reporting (e.g., Abstract, line 93, and line 115), and we have a specific paragraph (lines 125 – 132) acknowledging the limitations of our study.

#-----

I would also suggest citing some wider literature related to replicability and signal to noise ratio, as the authors are mainly citing their own papers and some replication projects and not many related papers are cited.

7REPLY 7: We would like to clarify that, as far as we know, except for the methods developed by our authors (EVZ and NI), few, if any, methods can estimate replicability without conducting a real replication project. Therefore, it is reasonable that we have cited our own papers. The reason why we have not cited many papers is due to the limited number of references required by the journal.

#-----

As the authors state, their study cannot replace substitute a true, collaborative replication project, and I am unsure of the real contribution this makes to the literature as it is already well known that the reproducibility is considerably lower than the acceptable rate in most fields (see some of the replication projects the manuscript is citing) and this manuscript provides a highly uncertain estimate of the replicability.

REPLY 8: As accurately pointed out by Reviewer 1 (see Reply 1), there are many difficulties in conducting large-scale, direct replications, and our computer simulation using available data fills an important gap. For example, only 50 of the 193 planned experiments (26%) were completed in the Reproducibility Project: Cancer Biology due to various barriers, including lack of transparency in the original methodology, failures to share original data, reagents, and other materials, and methodological challenges encountered during the execution of the replication experiments (10.7554/eLife.67995).

We also note that existing collaborative replication projects in other fields have severe flaws, although there are preferred. For example, all existing collaborative replication projects used extremely small samples (limited scope). Therefore, the replicability estimates from these replication projects cannot represent the replicability of the whole field. The very first replication project (Reproducibility Project: Psychology), published in Science, only involved 97 effect sizes (10.1126/science.aaf0918). The Science paper about the replication project in economics only involved 18 effect sizes (Experimental Economics Replication Project). From this point of view, we could argue that our method is better than the existing replication projects since our study has 88,218 effect sizes, which allows us to get at least an approximate idea of the state of the whole field, and is more likely to represent the whole field, although as we already mentioned real replications are extremely valuable.

Final Decision Letter:

29th July 2024

8Dear Dr Yang,

We are pleased to inform you that your Brief Communication entitled "A large-scale in silico replication of ecological and evolutionary studies", has now been accepted for publication in *Nature Ecology & Evolution*.

Over the next few weeks, your paper will be copyedited to ensure that it conforms to *Nature Ecology and Evolution* style. Once your paper is typeset, you will receive an email with a link to choose the appropriate publishing options for your paper and our Author Services team will be in touch regarding any additional information that may be required

Due to the importance of these deadlines, we ask you please us know now whether you will be difficult to contact over the next month. If this is the case, we ask you provide us with the contact information (email, phone and fax) of someone who will be able to check the proofs on your behalf, and who will be available to address any last-minute problems . Once your paper has been scheduled for online publication, the Nature press office will be in touch to confirm the details.

Acceptance of your manuscript is conditional on all authors' agreement with our publication policies (see www.nature.com/authors/policies/index.html). In particular your manuscript must not be published elsewhere and there must be no announcement of the work to any media outlet until the publication date (the day on which it is uploaded onto our web site).

Please note that *Nature Ecology & Evolution* is a Transformative Journal (TJ). Authors may publish their research with us through the traditional subscription access route or make their paper immediately open access through payment of an article-processing charge (APC). Authors will not be required to make a final decision about access to their article until it has been accepted. Find out more about Transformative Journals

Authors may need to take specific actions to achieve compliance with funder and institutional open access mandates. If your research is supported by a funder that requires immediate open access (e.g. according to Plan S principles) then you should select the gold OA route, and we will direct you to the compliant route where possible. For authors selecting the subscription publication route, the journal's

9standard licensing terms will need to be accepted, including <https://www.nature.com/nature-portfolio/editorial-policies/self-archiving-and-license-to-publish>. Those licensing terms will supersede any other terms that the author or any third party may assert apply to any version of the manuscript.

We welcome the submission of potential cover material (including a short caption of around 40 words) related to your manuscript; suggestions should be sent to Nature Ecology & Evolution as electronic files (the image should be 300 dpi at 210 x 297 mm in either TIFF or JPEG format). Please note that such pictures should be selected more for their aesthetic appeal than for their scientific content, and that colour images work better than black and white or grayscale images. Please do not try to design a cover with the Nature Ecology & Evolution logo etc., and please do not submit composites of images related to your work. I am sure you will understand that we cannot make any promise as to whether any of your suggestions might be selected for the cover of the journal.

You can generate the link yourself when you receive your article DOI by entering it here: <http://authors.springernature.com/share>.

[REDACTED]

P.S. Click on the following link if you would like to recommend Nature Ecology & Evolution to your librarian <http://www.nature.com/subscriptions/recommend.html#forms>

** Visit the Springer Nature Editorial and Publishing website at www.springernature.com/editorial-and-publishing-jobs for more information about our career opportunities. If you have any questions please click here.**